# Polycaprolactone Composite Micro/Nanofibrous Material as an Alternative to Restricted Access Media for Direct Extraction and Separation of Non-Steroidal Anti-Inflammatory Drugs from Human Serum Using Column-Switching Chromatography

**DOI:** 10.3390/nano11102669

**Published:** 2021-10-12

**Authors:** Hedvika Raabová, Lucie Chocholoušová Havlíková, Jakub Erben, Jiří Chvojka, František Švec, Dalibor Šatínský

**Affiliations:** 1The Department of Analytical Chemistry, Faculty of Pharmacy, Charles University, Akademika Heyrovského 1203, 50005 Hradec Králové, Czech Republic; raabovah@faf.cuni.cz (H.R.); havlikova@faf.cuni.cz (L.C.H.); svecfr@faf.cuni.cz (F.Š.); 2The Department of Nonwovens and Nanofibrous Materials, Faculty of Textile Engineering, The Technical University of Liberec, Studentská 1402/2, 46001 Liberec, Czech Republic; erben.jaakub@gmail.com (J.E.); jiri.chvojka@tul.cz (J.C.)

**Keywords:** restricted access media, nanofibers, microfibers, on-line extraction, biological samples, column-switching chromatography

## Abstract

Application of the poly-ɛ-caprolactone composite sorbent consisting of the micro- and nanometer fibers for the on-line extraction of non-steroidal anti-inflammatory drugs from a biological matrix has been introduced. A 100 μL human serum sample spiked with ketoprofen, naproxen, sodium diclofenac, and indomethacin was directly injected in the extraction cartridge filled with the poly-ɛ-caprolactone composite sorbent. This cartridge was coupled with a chromatographic instrument via a six-port switching valve allowing the analyte extraction and separation within a single analytical run. The 1.5 min long extraction step isolated the analytes from the proteinaceous matrix was followed by their 13 min HPLC separation using Ascentis Express RP-Amide (100 × 4.6 mm, 5 µm) column. The recovery of all analytes from human serum tested at three concentration levels ranged from 70.1% to 118.7%. The matrix calibrations were carried out in the range 50 to 20,000 ng mL^−1^ with correlation coefficients exceeding 0.996. The detection limit was 15 ng mL^−1^, and the limit of quantification corresponded to 50 ng mL^−1^. The developed method was validated and successfully applied for the sodium diclofenac determination in real patient serum. Our study confirmed the ability of the poly-ɛ-caprolactone composite sorbent to remove the proteins from the biological matrix, thus serving as an alternative to the application of restricted-access media.

## 1. Introduction

Non-steroidal anti-inflammatory drugs (NSAID) are widely used in pain, osteoarthritis, and rheumatoid arthritis treatment. A mechanism of their action lies in the inhibition of cyclo-oxygenase enzyme, an enzyme serving prostaglandin biosynthesis. These drugs play a crucial role in the suppression of the inflammatory response of an organism. The reduction in prostaglandin levels results in antipyretics, analgesics, and anti-inflammatory activity of NSAID. The most widely used NSAIDs in clinical practice are acetylsalicylic acid, paracetamol, indomethacin, ibuprofen, ketoprofen, sodium diclofenac, flurbiprofen, mefenamic acid, piroxicam, and nabumeton. Monitoring of NSAID in body fluid is essential for toxicological and pharmacokinetics studies. Their determination as a part of rational pharmacotherapy that aims to reduce risks and achieve disease treatment goals is less common. The levels are specifically monitored either in patients with an impaired renal function when the insufficient excretion occurs or when non-compliance is suspected [1,2]. The pharmaceutical industry is another sector where NSAID are determined in pharmaceutical formulations to ensure safety and effectiveness of drugs [3].

Numerous methods for NSAID determination in biological fluids including gas chromatography [4,5,6], capillary electrophoresis [7,8,9], and liquid chromatography [10,11] were developed over the past years. Although the analytical instruments were designed to provide increasingly faster and more sensitive analysis, the treatment of the biological matrixes prior to the instrumental analysis remains the bottleneck in the drug level monitoring [12]. The NSAID are usually determined in plasma, where the direct analysis of untreated matrix can lead to lower sensitivity, unprecise results, and in the worst case to irreversible damage of the chromatography column caused by protein precipitation. Thus, the sample pretreatment step cannot be omitted. This step is mainly based on time-consuming, laborious, and error-prone methods, such as protein precipitation, solid-phase extraction, and liquid–liquid extraction that can be potentially hazardous for laboratory staff [13]. Several approaches have been developed to overcome these shortcomings. The emphasis has been primarily placed on higher speed, automation, reduced consumption of organic solvents, smaller sample volumes, and increased selectivity [13,14]. Restricted access media (RAM) represent one of the applicable methods [15].

RAM sorbents separate the low molecular analytes from macromolecular interferences mostly via size exclusion effect. This analyte extraction and macromolecules removal occurs simultaneously after the injection of the untreated biological fluids [15]. The sample preparation is simplified and the analysis time is shortened while maintaining the desired extraction efficiency and avoiding column clogging [16]. The extraction mechanism of the original RAM is primarily based on their two surfaces. The outer surface is covered by hydrophilic groups avoiding access of macromolecules in the material, and the hydrophobic functional groups on the inner pore surface that are responsible for analyte retention. Additionally, the pores in the sorbent material contribute to the matrix removal by the size-exclusion mechanism [17].

The potential of RAM can be adequately utilized in on-line liquid chromatography extractions. The RAM cartridges are most often coupled with a column-switching chromatography system [16]. This setup was reported several times for NSAID analysis [18,19]. The main component of this system is a double position six-port selection valve that redirects the mobile phases between both extraction cartridge and analytical column. The extraction begins after the sample is injected in the cartridge. The undesired matrix components are removed from the analytes using the washing mobile phase. The column-switching valve then changes its position to complete the extraction step, and the analytes are eluted in the analytical column where they are separated using the separation mobile phase. The washing mobile is meanwhile transferred in the waste container. This process can be carried out either in a straight-flush or in a back-flush mode differing in the way the analytes are eluted from the extraction cartridge. They are injected and eluted from the extraction cartridge in the same direction as washing mobile phase flow in the former approach while the mobile phase elutes the analytes in the opposite direction to the sample injection in the latter [20].

Advanced approaches combine the RAM concept with the molecularly imprinted polymers, magnetism, or supramolecular solvents to enhance selectivity and specificity of the analyte extraction [21,22,23]. The use of nanomaterials is also included in these developments. For example, application of the restricted access nanotubes was reported [24]. We demonstrated first nanofibers as an alternative to RAM elsewhere [25]. We confirmed that the composite material produced from poly-ε-caprolactone exhibited satisfactory efficiency in extraction of parabens from human serum and bovine milk. The idea of using nanofibers as an alternative to RAM emerged as a result of their characteristics. The nanofibers have a large surface area to volume ratio enabling them to capture considerable quantities of low molecular weight analytes. In contrast, we speculated that the macromolecules are not retained because of the curvature of the fibers. The macromolecules appeared not to be flexible enough to be attached at multiple points to the highly curved nanofiber for their sufficient retention. This is not an issue with the low-molecular weight analytes [25]. Additionally, the large spaces between the fibers support the macromolecule passage through the sorbent to the waste.

We described the advantages of material combining poly-ɛ-caprolactone micro- and nanofibers (micro/nano PCL) for the on-line extraction of analytes from milk and serum matrixes in our previous study [25]. We confirmed the protein removal capability, good mechanical stability in the extraction/HPLC system, and re-usability of this sorbent for more than 300 analyses. The current study extends exploration of the potential of micro/nano PCL material in therapeutic drug monitoring and a method for NSAID determination in human serum samples was developed. This method was validated according to the International Council for Harmonization guideline and used for the determination of diclofenac content in a real patient serum.

## 2. Materials and Methods

### 2.1. Reagents and Materials

Ketoprofen (≥98.0%, KET), naproxen (≥98.0%, NAP), sodium diclofenac (≥98.5%, DCF), and indomethacin (≥98.5%, IND) were purchased from Sigma-Aldrich (Darmstadt, Germany) and used as model analytes. Phosphoric acid (Honeywell, Morris Plains, NJ, USA), acetonitrile (ACN), and methanol (MeOH) (VWR, Paris, France) served as mobile phase modifiers. The standard solutions were acidified by acetic acid from Penta (Prague, Czech Republic). The proteinaceous matrix was prepared by dissolution of lyophilized human serum Lyo hum N (Erba Lachema, Brno, Czech Republic). Water was purified by Millipore Milli-Q Direct Water Purification System from Merck (Darmstadt, Germany). The composite poly-ε-caprolactone sorbent was fabricated at the Technical University of Liberec from poly-ε-caprolactone (Mw 43.000) provided by Polysciences (Heddesheim, Germany), and chloroform (≥97%) and ethanol (99.97%) bought from Penta Chemicals (Pardubice, Czech Republic).

### 2.2. Instrumentation

#### 2.2.1. Column-Switching Chromatography System

A Shimadzu Prominence instrument (Shimadzu Corporation, Kyoto, Japan) equipped with three LC-20AD Prominence pumps, a DGU-20A Prominence on-line degasser, a SIL 20AC Prominence autosampler, a CTO-20AC Prominence column oven, an SPD-M20A Prominence UV/VIS photodiode array detector, and a CBM-20A Prominence system controller was used for the separation and analysis. Switching between the analytical column and the extraction cartridge was provided via the six-port high-pressure flow line switching valve FCV-12AH. LC Solution software (version 5.97, Shimadzu Corporation, Kyoto, Japan) controlled the instrument. The separation of NSAID extracted from the human serum was carried out on the Ascentis Express RP-Amide (100 × 4.6 mm, 5 µm) analytical column combined with the RP-Amide (5 × 4.6 mm, 2.7 µm) guard column (Sigma-Aldrich, Darmstadt, Germany). The system configuration in extraction and separation mode is depicted in the Figure 1.

#### 2.2.2. Meltblown and Electrospun Fibers

Poly-ɛ-caprolactone nanofibers and microfibers were fabricated using a process described in Appendix A and was firstly reported elsewhere [26,27]. The meltblown equipment (Laboratory equipment J&M Laboratories, Ashland, OH, USA) in combination with DC electrospinning lab-made system based on a multi-needle spinner was used for the preparation of the fibers. The production equipment consisting of both electrospinning and meltblown systems is depicted in Appendix A. A scanning electron microscope VEGA 3 (Tescan, Brno, Czech Republic) was used for the imaging of the fibrous sorbent. The morphology of micro/nanofibrous material is shown in Appendix A.

### 2.3. Preparation of Extraction Cartridge

A commercially available PEEK cartridge 4.6 × 10 mm (Merck, Darmstadt, Germany) was manually filled with 44 mg of the composite micro/nanofibrous PCL. The details of extraction cartridge preparation and schematic of filling are presented in Appendix A. The cartridge in a plastic holder was then placed in the chromatographic system and washed with the mobile phase containing 10% ACN in 0.085% (*v*/*v*) aqueous phosphoric acid for 15 min and neat ACN was then injected six times. This washing was carried out to ensure complete removal of impurities that could remain in fibers after their production.

### 2.4. Preparation of Standard and Matrix Solutions

The stock KET, NAP, DCF, and IND solutions were prepared at a concentration of 1 mg mL^−1^ by dissolving the individual standard substances in MeOH. These solutions were stored in the dark at 4 °C. The mix stock solution containing 0.2 mg mL^−1^ NSAID was prepared by mixing the stock solutions. An appropriate volume of the mix stock solution was diluted with a water–acetonitrile–acetic acid solution (50/48/2, *v*/*v*/*v*) to obtain standards solutions at seven concentration levels of 50; 100; 500; 1000; 5000; 10,000; and 20,000 ng mL^−1^. These solutions were used for the plotting the calibration curve. Commercially available lyophilized human serum was reconstituted according to manufacturer instructions and then diluted ten times with 20% aqueous ACN. This serum solution was spiked with the mix stock solution to obtain matrix solutions at concentrations of 50; 100; 500; 1000; 5000; 10,000; and 20,000 ng mL^−1^ and these solutions were centrifuged at 14,000× *g* rpm (21,578× *g*) for 15 min. The supernatant was then injected in the chromatographic system and the matrix calibration curve plotted. Matrix solutions were prepared fresh daily.

### 2.5. Real Sample

A real human serum was obtained from a patient administered with 250 mL continual intravenous infusion containing 75 mg of sodium diclofenac. Blood sampling was carried out in the Department of Clinical Biochemistry and Diagnostics of the University Hospital Hradec Králové. This real life serum was handled the same way as the matrix working solutions. First, it was diluted 10 times with 20% aqueous ACN and then centrifuged at 14,000× *g* rpm (21,578× *g*) for 15 min.

### 2.6. Analytical Method

The analytical run started with the extraction step when 100 µL sample, standard or matrix solution, was injected in the extraction cartridge filled with micro/nano PCL. The NSAID were captured by the sorbent while the proteins and other potentially interfering macromolecular substances were removed by the washing with the mobile phase composed of 10% ACN in 0.085% aqueous H_3_PO_4_. Simultaneously, the analytical column was conditioned using the 30% ACN in 0.085% aqueous H_3_PO_4_. The column-switching valve redirected this separation mobile phase after 1.5 min in the extraction cartridge. Hereby, the analytes were eluted in the analytical column and separated using the ACN gradient increasing the ACN percentage to 45% in 2.5 min and this mobile phase was pumped through the column for another 1.5 min. Then, the percentage was ramped to 55% in 4 min and to 75% in 0.5 min. Finally, the ACN percentage decreased to the initial concentration in 1 min at which the analytical column was re-equilibrated for 3 min. The flow rate of both mobile phases was held on 1 mL min^−1^. The total analysis time was 15 min. The separated analytes were detected using the diode array detector. KET, DCF, and IND were monitored at 270 nm while NAP at 232 nm. The column was held at 20 °C since the micro/nano PCL dissolve in ACN at higher temperatures [26].

## 3. Results and Discussion

### 3.1. Optimization of Chromatographic Separation

We tested ACN and MeOH as organic components of the mobile phase with ACN produced peaks with a better symmetry. The NSAID drugs are weak acids. Therefore, 0.085% aqueous H_3_PO_4_ solution pH 2.6 was used as the aqueous part of the mobile phase to increase the retention in the column. We tested two analytical columns Ascentis Express RP-Amide (100 × 4.6 mm, 5 µm) and Ascentis Express F5 (100 × 4.6 mm, 5 µm) for the separations. Both columns use the core-shell particles technology improving the efficiency and reducing the flow resistance in column switching system. However, they are more susceptible to impurities from proteinaceous samples resulting in an increase in back pressure and eventually column clogging. Therefore, the Ascentis Express (5 × 4.6 mm, 2.7 µm) guard columns packed with the same stationary phase were inserted in the system to protect the analytical column. The Ascentis Express RP-Amide analytical column enabled a better separation of the analytes. Ascentis Express F5 stationary phase was not well suited because the extensive peak broadening and coelution of ketoprofen-naproxen and sodium diclofenac-indomethacin pairs as presented in Figure 2.

### 3.2. Optimization of On-Line Extraction

We reported elsewhere [25] that the removal of proteins from nanofibrous sorbent using an aqueous mobile phase occurs within the first minute. This finding was confirmed by monitoring elution at a wavelength of 280 nm that is the absorption maximum for proteins. Human serum contains in addition to proteins also more lipophilic ballast substances, such as, for example, lipoproteins and vitamins, that are difficult to remove from the system since aqueous washing is insufficient. In contrast, a higher concentration of organic solvents in the washing mobile phase and excessive washing lead to undesirable losses of desired analytes. Therefore, the composition of the washing mobile phase and the duration of extraction step had to be optimized to remove most of the lipophilic ballast molecules without any loss of the analytes. Removal of these compounds using the mobile phases containing ACN concentration of 5%, 10%, 15%, and 20% were tested. The recovery of adsorbed analytes on nanofibers decreased with the rising ACN concentration. However, the effect of this process on the removal of ballast substances was negligible. This is why we used a mixture comprising 10% ACN in 0.085% aqueous phosphoric acid. This washing mobile phase was the best compromise between analyte loss and ballast removal. The duration of the extraction step was studied applying the similar approach. The peak of matrix impurities was observed in the chromatogram at 270 nm after 1 min washing. Extension of the washing time by 30 s resulted in a reduction in the area of the matrix ballast peaks without reducing the peak areas of analytes. No further improvement was observed after further increasing the extraction time to 2 min. Thus, the extraction time was finally held at 1.5 min. Chromatogram of standard solution and spiked serum under the optimized conditions of extraction and separation is shown in Figure 3.

### 3.3. Extraction Efficiency

The analyte recovery from the human serum matrix was compared to the recovery obtained using extraction from standard solutions and expressed as a percentage value considering the standard solution being 100%. Figure 4 demonstrates that at levels up to 100 ng mL^−1^, the analytes are extracted from the serum with an efficiency comparable to that found for the standard solutions. KET is an exception since its recovery exceeded 100% at almost all concentration levels. It can be caused by coelution with a matrix interference that increased the total peak area. This finding reduces the accuracy of the method for KET. However, the RSD values ranging from 1.16% to 0.017% and the precision is not affected.

### 3.4. Optimization of Injected Sample Volume

Different injection volumes were tested to achieve better sensitivity using UV detection and to simultaneously determine the protein removing capacity of the extraction cartridge. The proteins were completely removed in less than 1 min after the injection volumes 10, 25, and 50 µL. The cleaning of up of 100 µL centrifuged serum was achieved in 1.5 min. A further increase in the injection volume and simultaneous extension of the washing phase resulted in larger analyte loss. Therefore, 100 µL serum was designated as the maximal injection volume for the 10 × 4.6 mm i.d. extraction cartridge.

### 3.5. Validation

The method was validated with respect to the linearity, precision, accuracy, selectivity, and sensitivity following the ICH Q2 R1 guideline to evaluate the reliability of the results [28]. The validation parameters were chosen to primarily confirm micro/nanofibrous PCL applicability for the NSAID extraction since this sorbent is not standardized and commonly used. The ICH Q2 R1 protocol meets these requirements better than the M10 guideline that is usually used for the validation of bioanalytical methods. The chromatographic system suitability test was carried out confirm the suitability of the HPLC instrument for NSAIDs analysis. Mean values and standard deviations of the retention time, peak capacity, symmetry factor, resolution, and repeatability of the analytical run were calculated from results of six injections of standard solutions and evaluated according to the European Pharmacopoeia recommendations. The results are presented in the Table 1.

#### Linearity, Accuracy, Precision, and Limits of Detection and Quantification

All samples were measured in triplicate using the optimized conditions. The calibration curves were established for standard and serum matrix solutions at concentration levels of 50; 100; 500; 1000; 2000; 10,000; and 20,000 ng mL^−1^. The linear relationship between the NSAID quantity and peak area was confirmed for KET, DCF, and IND in the concentration range 50 to 20,000 ng mL^−1^. The calibration curve for NAP was linear for a range 50 to 10,000 ng mL^−1^. The correlation coefficient for each drug was exceeded 0.996. Standard and matrix matched calibration curves including regression equations are demonstrated in Appendix A.

The accuracy was determined via the recovery study carried out for all the drugs in the human serum matrix. The recovery was calculated as a comparison of peak areas of NSAID in standard and serum matrix solutions at three concentration levels of 100; 1000 and 10,000 ng mL^−1^ measured in triplicate. The results of recovery also determined the extraction efficiency for each drug.

The serum solutions were measured six times at three concentration levels of 100; 1000; and 10,000 ng mL^−1^ for intra-day precision determination. The results expressed as RSD (%) were determined in a range of 7.89% to 10.24% for a concentration level of 100 ng mL^−1^, 0.17% to 0.90% for a concentration level of 1000 ng mL^−1^, and 0.09% to 0.46% for a concentration level of 10,000 ng mL^−1^. The inter-day precision was calculated for three consecutive measurements of matrix spiked with 1000 ng mL^−1^ NSAID in three days. The results expressed as RSD (%) were in a value range of 3.62 and 7.88%. The lowest concentration of the calibration curve equaled to 50 ng mL^−1^ (10σ) was established as a limit of quantification (LOQ). The limit of detection 15 ng mL^−1^ (LOD) was calculated from the LOQ value as a tree-folds (3σ) variation. The results of the validation are summarized in the Table 2. These results are comparable to those obtained using other methods described for diclofenac determination, including column-switching [18], protein precipitation [29], and SPE via commercial sorbent [30] applied as sample preparation procedures.

### 3.6. Reusability of the Extraction Cartridge

The manually packed extraction cartridge was used during the entire experiments corresponding to the analysis of more than 100 serum samples. Neither extraction efficiency nor the back pressure of PCL composite sorbent had changed during these experiments. Thus, our composite micro/nanofibrous sorbent is better suited for the extensive use in high-pressure liquid chromatography systems than nanofibers reported elsewhere [31]. The microfibers represent a stable scaffold resistant to the high pressure while, simultaneously, the large surface area to volume ratio of the nanofibers contributes to the high extraction efficiency. Moreover, the cotton-like texture of micro/nano composite material is easier to manually fill into the cartridge. The durability of our cartridge paralleled our previous study [20] where we applied 200 injections of 200 µL serum in commercial RAM LiChrospher RP-18 ADS column without observing a decrease in extraction efficiency.

### 3.7. Analysis of Real Sample

The real patient serum samples after continual intravenous infusion containing 75 mg sodium diclofenac were handled as a serum matrix and the content of diclofenac determined using our method. The diclofenac peak was recognized based on its retention time and UV spectrum. The calculated concentration of diclofenac was 32.57 ng mL^−1^ in ten times diluted serum corresponding to 325.7 ng mL^−1^ in the original patient serum. This result confirmed that the extraction using nanofibrous sorbent is enough sensitive to handle the real samples. The chromatogram of the patient serum is shown in Figure 5.

## 4. Conclusions

Our study built on previous extraction experiments with micro/nano PCL that had promising properties enabling the direct extraction of analytes from proteinaceous matrix. We tested the micro/nano PCL fibers as the extraction sorbent for on-line solid-phase extraction of NSAID. After the injection of 100 μL human serum matrix, the sorbent enabled removal of the proteins and the majority of other macromolecular interferences within 1.5 min. The analyte recovery from the matrix was comparable to that obtained with the extraction from solutions of standards. This demonstrated that micro/nano PCL is a promising sorbent for KET, NAP, DCF, and IND even from the proteinaceous matrixes. The applicability of micro/nano PCL sorbent for the therapeutic drug monitoring was explored using a real sample containing diclofenac. The extraction provided good sensitivity even at low concentrations typical of biological samples and purified them sufficiently. Based on our results, we expect that the nano/micro fibrous PCL sorbent has a potential to improve the bioanalytical methods for sample pretreatment in terms of higher effectiveness at a lower costs compared to commercial RAM. This makes it a valuable tool for the on-line extraction/chromatography methods developed for the therapeutic drug monitoring enabling fast, safe, and precise analysis.

## Figures and Tables

**Figure 1 nanomaterials-11-02669-f001:**
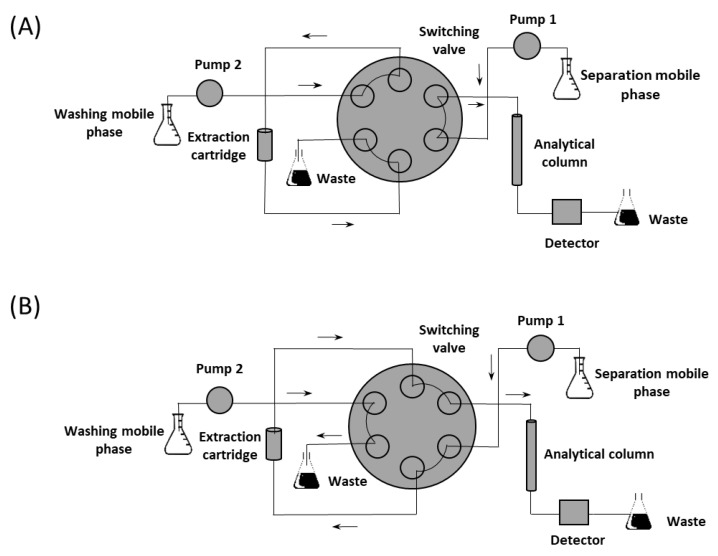
Column-switching chromatographic system in (**A**) on-line extraction and (**B**) on-line extraction-separation mode.

**Figure 2 nanomaterials-11-02669-f002:**
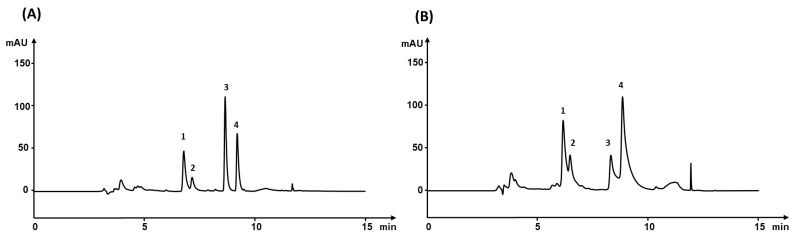
Chromatograms of standard solution spiked with 5000 ng mL^−1^ NSAID separated on (**A**) Ascentis Express RP-Amide column and (**B**) Ascentis Express F5 column at a wavelength 270 nm. 1—ketoprofen, 2—naproxen, 3—sodium diclofenac, 4—indomethacin.

**Figure 3 nanomaterials-11-02669-f003:**
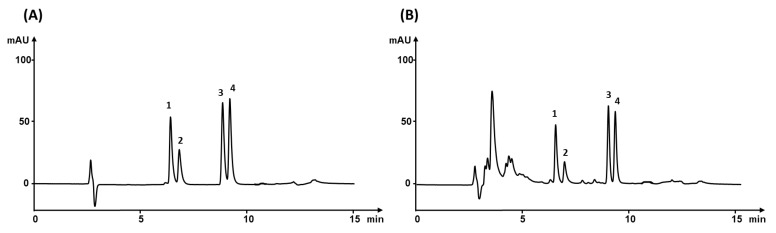
Chromatograms of (**A**) standard solution and (**B**) matrix solution spiked with 1000 ng mL^−1^ NSAID at a wavelength 270 nm. 1—ketoprofen, 2—naproxen, 3—sodium diclofenac, 4—indomethacin.

**Figure 4 nanomaterials-11-02669-f004:**
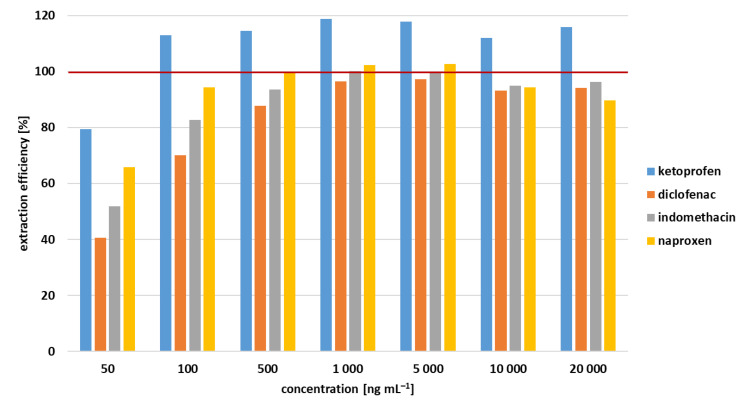
Extraction efficiency of micro/nano PCL sorbent expressed as a percentual recovery of NSAID in human serum.

**Figure 5 nanomaterials-11-02669-f005:**
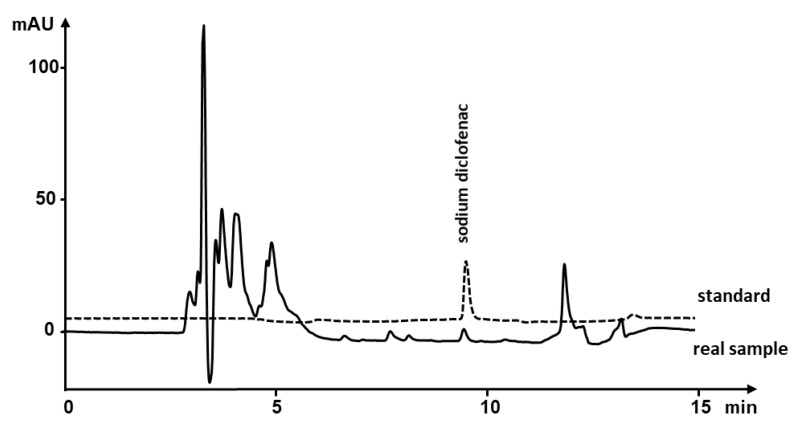
Standard solution spiked with 50 ng mL^−1^ diclofenac and real patient human serum containing the same drug. The concentration of diclofenac in real sample was equal to 32.57 ng mL^−1^ in ten times diluted serum.

**Table 1 nanomaterials-11-02669-t001:** HPLC system suitability parameters. Sample injection was performed in six replicates.

	Ketoprofen	Naproxen	Sodium Diclofenac	Indomethacin
Retention time [min]	6.41	6.82	9.18	10.49
Retention time repeatability, RSD [%]	0.03	0.03	0.02	0.02
Repeatability of peak areas, RSD [%]	0.85	0.49	0.21	0.56
Peak resolution	1.82	10.32	5.99	
Peak symmetry	1.80	1.63	2.00	1.60
Peak capacity	14.03	13.18	21.65	18.80

**Table 2 nanomaterials-11-02669-t002:** HPLC method validation results.

	Ketoprofen		Naproxen	Sodium Diclofenac	Indomethacin
Standard calibration range [ng mL^−1^] ^1^	50–20,000		50–10,000	50–20,000	50–20,000
Correlation coefficient	0.9997		0.9996	0.9999	0.9998
Matrix calibration range [ng mL^−1^] ^2^	50–20,000 ^1^		50–10,000 ^2^	50–20,000 ^1^	50–20,000 ^1^
Correlation coefficient	0.9999		0.9960	0.9999	0.9999
Intra-day precision [%] ^3^	c_1_	8.54	7.89	10.24	8.37
	c_2_	0.60	0.90	0.38	0.17
	c_3_	0.46	0.29	0.09	0.21
Inter-day precision [%] ^4^		6.79	3.62	7.44	7.88
Accuracy [%] ^5^	c_1_	112.91 ± 4.50	94.31 ± 4.09	70.07 ± 6.20	82.76 ± 6.96
	c_2_	118.66 ± 0.84	102.28 ± 0.46	96.42 ± 0.15	100.06 ± 0.22
	c_3_	111.87 ± 0.70	94.24 ± 0.23	93.12 ± 0.83	94.81 ± 0.20
LOD [ng mL^−1^] ^6^		2.49	24.40	3.58	26.57
LOQ [ng mL^−1^] ^7^		8.29	81.34	11.92	88.57

^1^ Each concentration of calibration standard was measured in triplicate; ^2^ Each concentration of matrix calibration standard was measured in triplicate. Calibration curves were linear for (1) ketoprofen, diclofenac sodium, and indomethacin at seven concentration levels, (2) naproxen at six concentration levels; ^3^ Relative standard deviation (RSD) was calculated from six injections of the matrix solutions spiked with analytes at concentration levels c_1_ = 100 ng mL^−1^, c_2_ = 1000 ng mL^−1^ and c_3_ = 10,000 ng mL^−1^; ^4^ Relative standard deviation (RSD) was calculated from the average of three injection of matrix solutions spiked with analytes at concentration level c = 1000 ng mL^−1^ for three days; ^5^ Accuracy was determined as a method recovery of matrix solutions at concentration levels c_1_ = 100 ng mL^−1^, c_2_ = 1000 ng mL^−1^ and c_3_ = 10 000 ng mL^−1^ measured in triplicate.; ^6^ Limit of detection (LOD) was calculated from the signal-to-noise ratio in a 3-fold (3σ) variation; ^7^ Limit of quantification (LOQ) was calculated from the signal-to-noise ratio in a 10-fold (10σ) variation.

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
