# Peer review of "Polycaprolactone Composite Micro/Nanofibrous Material as an Alternative to Restricted Access Media for Direct Extraction and Separation of Non-Steroidal Anti-Inflammatory Drugs from Human Serum Using Column-Switching Chromatography"

_nanomaterials, 2021, doi:10.3390/nano11102669_

Round 1
Reviewer 1 Report
In my opinion, this work was properly designed and the manuscript presented has been well prepared. I only wonder, as Nanomaterials is the target journal, less information has been provided with regard to the structure of the sorbent, and this research is a direct application case to the sorbent. So what's the novelty of the current work, especially compared with the conventional approaches.
Author Response
Thank you for your encouraging words. Regarding your doubts about a journal choice, we believe, our manuscript meets the aims and scope of Nanomaterials and its special issue - Nanomaterials for Biomedical and Biotechnological Applications. The study builds on our previous knowledge [ref: Raabova, H.; Hakova, M.; Havlikova, L. C.; Erben, J.; Chvojka, J.; Solich, P.; Svec, F.; Satinsky, D., Poly-epsilon-caprolactone Nanofibrous Polymers: A Simple Alternative to Restricted Access Media for Extraction of Small Molecules from Biological Matrixes. Analytical Chemistry 2020, 92 (10), 6801-6805.] and develops an idea of the new application of nanofibers in analysis of biofluids. The innovative use of materials in nanoscale belongs to the topics covered in the Nanomaterials. However, you are right that the production of nanomaterials itself has been summarized very briefly in our article, so we have included its detailed description together with the characterization of the fibers in the Supporting Information.
Reviewer 2 Report
This study focuses on a research of extraction and separation of NSAID from human serum using a technology of PLA nanofibers assembled column-switching chromatography. Overall, the manuscript was well written in the section of extraction and separation of NSAID. However, the scope of the NANOMATERIALS journal focuses on materials. This study ignored the most important part regarding the materials, including material preparation and optimization. The authors should add this section and discuss the effect of the selection and the change of material preparation on the effectiveness of extraction and separation of NSAID. If this addition makes the length of manuscript pretty long, the authors should make a summary and put it to a supplementary information attached to the manuscript.
- The title may need to be further polished. It is confused.
- The first paragraph of the introduction should add a few examples of NSAID.
- The most important parts in this study are 2.2.2 meltblown and electrospun fibers and 2.3 preparation of extraction cartridge. The authors did not clearly address these steps. Specifically, it is not clear about how to prepare extraction cartridge. There are not quantitative information. For example, how much weigh of PCL fibers added to the PEEK cartridge? how compact were PCL fibers loaded in to the column? Fiber porosity or the porosity of the assembled cartridge? It seems many steps were missing.
- The size of PCL micro/nanofibers may need to be added to the section of Results and discussion. Did the size, the porosity and the loading capacity and others affect the effectiveness of extraction and separation? all these important points regarding the PCL fibers need be further addressed.
Author Response
We appreciate your critical point of view. You are definitely right, the used nanomaterials are not described properly. Therefore, the detailed fabrication description and characteristics of produced fibers were added as the Supporting Information.
1. The title may need to be further polished. It is confused.
Thank you for your opinion. We changed the title to better communicate the main idea of our research.
2. The first paragraph of the introduction should add a few examples of NSAID.
We have carefully considered your suggestion. The most widely used NSAIDs in clinical practice are acetylsalicylic acid, paracetamol, indomethacin, ibuprofen, ketoprofen, sodium diclofenac, flurbiprofen, mefenamic acid, piroxicam, and nabumeton.
3. The most important parts in this study are 2.2.2 meltblown and electrospun fibers and 2.3 preparation of extraction cartridge. The authors did not clearly address these steps. Specifically, it is not clear about how to prepare extraction cartridge. There are not quantitative information. For example, how much weigh of PCL fibers added to the PEEK cartridge? how compact were PCL fibers loaded in to the column? Fiber porosity or the porosity of the assembled cartridge? It seems many steps were missing.
We were more focused on the cartridge application itself and so we kept the material section brief in this manuscript. As we mentioned before, we have included the details about the nanofiber fabrication and morphology to the Supporting Information. We admit, it improves the intelligibility and fluency of the manuscript. A closer explanation of cartridge filling can be found in the Supporting Information as well.
Porosity in term of inter-fiber rooms are demonstrated in Supporting Information Figure S-2. No significantly changes in material structure were observed before and after using in high pressure system. Additionally, the amount of the fibers is not important in case of on-line extraction. We used only one cartridge fill completely with approximately 44 mg of the sorbent during the whole experiment. If we would prepare more cartridges, this weight should be maintained to reach the same extraction efficiency. The main requirement of the sufficient extraction is completely filled cartridge without void volumes.
4. The size of PCL micro/nanofibers may need to be added to the section of Results and discussion. Did the size, the porosity and the loading capacity and others affect the effectiveness of extraction and separation? all these important points regarding the PCL fibers need be further addressed.
Thank you for the question. The porosity of the fiber may be a confusing expression. As was already mentioned, the fibers itself are not porous. The fabricated material used as a sorbent in our study has a cotton-like structure. Therefore, the porosity can be only explained as the spaces between the single fibers. However, neither area nor size of these spaces in material are important to us. The cartridge must be fully filled with nanofibers to extract the samples properly. Therefore, the fibers are pressed into the cartridge which lead to compact sorbent without the void volumes. The loading capacity was sufficient to retain the highest concentration injected several times. Recovery values close to 100% confirmed the effectiveness of extraction sorbent. Separation was not affected because it depends on analytical chromatography column.
Reviewer 3 Report
The recoveries from an extraction method should be constant with the analyte concentration in the linear range of the method.
In the study of the efficiency of the proposed method (Figure 4), the recoveries of the analytes decrease significantly for low analyte concentrations (<500 ng / mL), and these concentrations are higher than the limit of quantification (50 ng / mL). If the recoveries depend on the concentration of the analytes, how can the concentration of the compounds in the range between 50 and 500 ng / mL be calculated?
In this work, the limit of quantification is the lowest cncentration prepared for the the calibration curve. This is a relative value and is not a calculated value. The quantification limits must be calculated statistically, and they are different for each compound. The sensitivity is not determined in the validation of the method.
Author Response
- The recoveries from an extraction method should be constant with the analyte concentration in the linear range of the method.
Yes, we confirmed it using matrix calibration range. The slopes were almost identical. The lower recovery values for sodium diclofenac and indomethacin are caused by its higher lipophilicity and thus higher rate of binding to proteins. This effect is more obvious for lower concentration.
- In the study of the efficiency of the proposed method (Figure 4), the recoveries of the analytes decrease significantly for low analyte concentrations (<500 ng / mL), and these concentrations are higher than the limit of quantification (50 ng / mL). If the recoveries depend on the concentration of the analytes, how can the concentration of the compounds in the range between 50 and 500 ng / mL be calculated?
Thank you for the inspiring question. The decreasing recovery for the concentration lower than 500 ng. mL-1 can be explained as partial interaction/binding with protein matrix that is more significant for lower concentrations. From our point of view, it is important to set those sensitivity limits. It simply tells us, not all the analytes are captured from the sample if their concentration is lower than 500 ng. mL-1. However, it does not mean the concentration cannot be calculated. For these cases, the matrix calibration is optimal solution of the problem. Therefore, the matrix calibration was evaluated in our study (table 2)
- In this work, the limit of quantification is the lowest concentration prepared for the the calibration curve. This is a relative value and is not a calculated value. The quantification limits must be calculated statistically, and they are different for each compound. The sensitivity is not determined in the validation of the method.
We appreciate your suggestion, the calculated LOQ are listed in the Table 2. The method sensitivity can be derived from these values. Nevertheless, all the values must be confirmed experimentally by multiple dilution of sample and reliable determination. Limit of quantification (LOQ) was calculated from the signal-to-noise ratio in a 10-fold (10σ) variation and LOD as (3σ). Values were added in Table 2.
Round 2
Reviewer 2 Report
The revised manuscript has address all the concerns given by the reviewers.
Author Response
The revised manuscript has address all the concerns given by the reviewers.
Thank you for the positive evaluation. We have no further comment.
Reviewer 3 Report
Please, can you explain what is the difference between the standard calibration and matrix calibration, since the linearity is the same in both cases and the concentration of the analyte can be below 500 ng mL-1?
The lower standard concentration is 50 ng mL-1 and in your opinion, the values below 500 ng mL-1 are not completely extracted because of the proteins, how do you eliminate the interference by the addition of 50 ng mL-1 of the analyte to the matrix?
Author Response
Please, can you explain what is the difference between the standard calibration and matrix calibration, since the linearity is the same in both cases and the concentration of the analyte can be below 500 ng mL-1?
Standard calibration solutions are prepared in organic solvent only. While matrix calibration solutions are prepared directly in biological matrix (spiking of blank human serum) to avoid unprecision caused by matrix effect (proteins, interferences, endogenous substances, etc.). If the matrix calibration is done, the evaluation of analyte concentrations is more reliable. Mainly in case of lower recovery values. Analyte shows same behavior in real samples and in spiked matrix. Linearity is based on the choice of the analyst and real conditions during the method validation and real sample concentrations. The most important factor is slope of calibration curve. It can show the presence of matrix effect and the source of uncertainty.
In case of different slopes of calibrations (matrix/standard) the better precision is reached using the matrix one. Therefore, both graphs, standard and matrix matched, of calibration of all analytes were added to Supporting info section.
The lower standard concentration is 50 ng mL-1 and in your opinion, the values below 500 ng mL-1 are not completely extracted because of the proteins, how do you eliminate the interference by the addition of 50 ng mL-1 of the analyte to the matrix?
You are true. The interference is not eliminated. The only way to reliable quantify the analytes concentration in real sample is the using matrix calibration curve. By this way, the negative effect of lower recovery or potential interferences will be eliminated. This is a standard procedure in validation of bioanalytical methods.